# Dietary Cottonseed Protein Substituting Fish Meal Induces Hepatic Ferroptosis Through SIRT1-YAP-TRFC Axis in *Micropterus salmoides*: Implications for Inflammatory Regulation and Liver Health

**DOI:** 10.3390/biology14070748

**Published:** 2025-06-23

**Authors:** Quanquan Cao, Ju Zhao, Xuefei Zhang, Laia Ribas, Haifeng Liu, Jun Jiang

**Affiliations:** 1College of Animal Science and Technology, Sichuan Agricultural University, Chengdu 611130, China; qq2312159@sicau.edu.cn (Q.C.); 2024102020@stu.sicau.edu.cn (J.Z.); 2020302169@stu.sicau.edu.cn (X.Z.); 2Institut de Ciències del Mar, Consejo Superior de Investigaciones Científicas (ICM-CSIC), 08003 Barcelona, Spain; lribas@icm.csic.es

**Keywords:** cottonseed protein, inflammatory regulation, ferroptosis, liver fibrosis, *M. salmoides*

## Abstract

This study investigated the effects of high-level cottonseed protein (CP) inclusion in diets on largemouth bass (*M. salmoides*). Replacing fish meal with CP beyond 24% impaired growth and liver health, elevating inflammation and ferroptosis markers. Excessive CP (60%) disrupted mitochondrial integrity, triggered oxidative stress, and activated ferroptosis via SIRT1-YAP-TRFC signaling, highlighting the need for balanced CP substitution in aquafeeds.

## 1. Introduction

Fish meal (FM) is a crucial high-quality protein source in aquafeeds, prized for its excellent palatability, high digestibility, and rich protein content [1]. However, the ecological and economic sustainability of FM-dependent aquaculture systems faces dual challenges: volatile market prices linked to fluctuating marine harvests and mounting environmental concerns over wild fishery depletion [2]. These constraints have intensified global efforts to identify sustainable plant-based protein alternatives that can partially or fully replace FM without compromising aquatic animal health [3].

Cottonseed protein (CP), a byproduct of cotton oil extraction, has emerged as a promising candidate for aquafeed formulation. Through advanced processing techniques that reduce crude fiber and free gossypol [4,5], CP now achieves protein content exceeding 60% (dry weight basis) with essential amino acid profiles comparable to soybean meal [6]. Successful FM substitution trials using CP have been documented across phylogenetically diverse species, including rainbow trout (*Oncorhynchus mykiss*) [7], juvenile large yellow croaker (*Larimichthys crocea*) [8], juvenile southern flounder (*Paralichthys lethostigma*) [9], pearl gentian grouper (*Epinephelus fuscoguttatus* ♀ × *E. lanceolatu* ♂) [10], black tiger shrimp (*Penaeus monodon*) [11], and common carp (*Cyprinus carpio*) [12,13]. Nevertheless, emerging evidence reveals threshold-dependent hepatotoxicity: 18% CP substitution induced hepatic lipid vacuolation in largemouth bass (*M. salmoides*) [14], while combined plant protein blends reduced antioxidant capacity (e.g., T-AOC and GSH-Px) in Japanese sea bass [15]. These findings underscore the critical need to delineate the molecular mechanisms underlying CP-induced hepatic pathology.

Mounting evidence implicates ferroptosis—an iron-dependent cell death pathway driven by lipid peroxidation—in chemical-induced liver injury [16]. This process is characterized by glutathione (GSH) depletion, mitochondrial cristae fragmentation, and transferrin receptor (TFRC)-mediated iron overload [17]. Notably, CP components exhibit dual pro-ferroptotic effects: gossypol directly depletes GSH through thiol conjugation [18], while CP hydrolysates downregulate sirtuin 1 (SIRT1) expression [19]. SIRT1, an NAD^+^-dependent deacetylase, modulates both ferroptosis and the Hippo/YAP signaling pathway [20]. The SIRT1-YAP axis further amplifies iron uptake through TFRC transactivation [21], creating a pathogenic cascade potentially central to CP hepatotoxicity.

Available evidence indicates that CP is a promising plant-based protein source for aquafeeds. However, excessive substitution of FM with CP has been associated with adverse effects, including hepatic structural damage, elevated ROS production, and oxidative stress. Given the economic and ecological significance of *M. salmoides* in Chinese freshwater aquaculture, evaluating the feasibility and potential risks of CP as an FM alternative is crucial. This study aimed to assess the effects of graded FM replacement with CP on growth performance and hepatic ferroptosis in *M. salmoides*.

## 2. Materials and Methods

### 2.1. Animal Ethics Statement

The present study followed the recommendations of Care and Use of Laboratory Animals in China, Animal Ethical and Welfare Committee of China Experimental Animal Society. All experimental procedures conducted in this study received approval from the Animal Care Advisory Committee of Sichuan Agricultural University, under permit number DKY-2018202027.

### 2.2. Experimental Diets and Feeding Management

In this study, CP was procured from Chenguang Biotechnology Group Kashgar Co., Ltd. (Kashgar, China). Diets were formulated with CP inclusion levels ranging from 7% to 35%, replacing FM protein at rates of 12%, 24%, 36%, 48%, and 60% (designated as CP12, CP24, CP36, CP48, and CP60, respectively). Specific lysine and methionine levels were adjusted according to established standards [22]. All ingredients were finely ground, pelletized into 4.0 mm floating pellets using a double-screw extruder, and stored at −20 °C until use. The formulation, nutrient composition, and chemical analysis of four isonitrogenous and isolipidic diets (crude protein, 52.0%; crude lipid, 8.0%) and nutrient levels are presented in Table 1.

The *M. salmoides* were sourced from a local aquafarm in Sichuan, China. Before the start of the experiments, the fish were fed a basal diet for two weeks to acclimate them to the experimental conditions. A total of 720 fish, with an average initial weight of 39.36 ± 0.10 g, were carefully selected and randomly distributed into 18 tanks (200 × 100 × 105 cm^3^). There were three replicates (tanks) for each treatment, with 40 fish per tank. The fish were fed ad libitum with experimental diets until they appeared satiated twice a day (at 7:00 AM and 7:00 PM) for a duration of 8 weeks. Fish were held under natural photoperiod condition throughout the feeding trial. Each tank received freshwater at a flow rate of 1.2 L per minute. The water temperature ranged from 26 ± 3 °C, and the dissolved oxygen level was maintained above 5 mg per liter. The pH was kept within the range of 7.1–7.3 and total ammonia nitrogen was <0.04 mg/L.

### 2.3. Sample Collection and Indicator Measurements

Prior to the feeding trial, samples from 40 fish in each treatment group were stored at −20 °C for proximate composition analysis to calculate nutrient retention. At the trial’s conclusion, the fish were fasted for 24 h, and individual tanks were counted and weighed to calculate performance indices. Following anesthesia with 0.01% MS-222, blood samples were collected from the tail veins of nine fish per group and centrifuged at 1000 g for 10 min at 4 °C to isolate serum for analysis. The fish were immediately dissected, and the visceral mass was removed and weighed. Tissues such as the liver, intestine, and stomach were then separated for further analysis. The following parameters were measured as previously described [23]: survival rate (SR), initial body weight (IBW), final body weight (FBW), percent weight gain (PWG), specific growth rate (SGR), feed intake (FI), feed efficiency (FE), relative gut length (RGL), intestosomatic index (ISI), viscerosomatic index (VSI), and hepatosomatic index (HSI). Additionally, a 1 cm^3^ central portion of the liver was fixed in a 4% paraformaldehyde solution for histological analysis. The remaining liver tissue was frozen in liquid nitrogen for subsequent RNA and protein extraction experiments.

### 2.4. Proximate Composition Analysis

The proximate composition of diets, as well as whole-body and dorsal muscle fish samples, was analyzed according to the procedures outlined by the Association of Official Analytical Chemists [24]. For moisture determination, diet samples were dried at 105 °C until reaching a constant weight, while fish samples underwent vacuum freeze-drying. A semi-automatic nitrogen analyzer (KDN-08A, Shanghai Xinjia Electron Co., Ltd. Shanghai, China) was employed to measure crude protein content via the Kjeldahl method. The Soxhlet extraction technique was utilized to quantify the crude lipid content of diets, and ash content was determined by incinerating diet samples in a muffle furnace at 550 °C.

### 2.5. Liver Haematological and Homogenate Parameters

Triglyceride (TG, A110-1-1), total cholesterol (TC, A111-1-1), alanine aminotransferase (ALT, C009-2-1), aspartate aminotransferase (AST, C010-2-1), alkaline phosphatase (AKP, A059-2-2), L-Hydroxyproline (HYP, A030-2-1), Glutathione S-transferase (GST, A004-1-1), Malondialdehyde (MDA, A003-1-2), Adenosine triphosphate (ATP, A095-1-1), and total protein (TP, A045-2-2) levels in serum and liver homogenate were determined using assay kits (Catalog No.A002-045, Nanjing Jiancheng Co., Nanjing, China) following the manufacturer’s protocols.

### 2.6. Hematoxylin and Eosin Staining

Liver tissues were fixed in 4% paraformaldehyde (Catalog No. P6148, Sigma-Aldrich, Saint Louis, MI, USA), embedded in paraffin (Leica, Wetzlar, Germany, Catalog No. 39601006), and cut into 4 μm thick sections. After dewaxing using different concentrations of xylene (Fisher Chemical, Pittsburgh, PA, USA, Catalog No. X5S-1) and ethanol, the nuclei were stained with hematoxylin solution (Sigma-Aldrich, Catalog No. HHS32), and the cytoplasm was stained with eosin solution (Sigma-Aldrich, Catalog No. HT110232). The sections were then dehydrated and made transparent with alcohol and xylene. Subsequently, the sections were sealed with neutral gel, and pathological changes were observed under a microscope (Olympus CX43, Olympus Corp. Tokyo, Japan).

### 2.7. Fe^2+^ Content Detection

The Fe^2+^ content in liver tissue was determined using a colorimetric method according to the instructions provided (E-BC-K773-M, Wuhan Elabscience Biotechnology Co., Ltd., Wuhan, China). Liver tissues were added to the ferrous kit buffer, homogenized thoroughly on ice using a homogenizer, then centrifuged at 10,000× *g* for 10 min and the supernatant was collected for subsequent assays. Finally, 300 μL of the reagent mixture was incubated with 150 μL of supernatant in a 1.5 mL EP tube for 10 min at 37 °C, and the optical density (OD) value was measured at 593 nm.

### 2.8. Molecular Docking

The small molecular structure of gossypol was downloaded from the PubChem database (https://pubchem.ncbi.nlm.nih.gov/, 6 June 2022). Hydrogenation and energy minimization were carried out using Chem3D, v19.0.0.22. The crystal structure of SIRT1 was obtained from the SIRT1 protein database (PDB code: 4L7B). Molecular docking of gossypol and SIRT1 was performed using AutoDock Vina v1.2.0 to determine the binding energy. Three-dimensional binding patterns of protein–small molecule complexes were analyzed and mapped using PyMOL v2.6.1, while Discovery Studio 2020 client was employed for further analysis and mapping of two-dimensional interaction patterns of protein–small molecule complexes.

### 2.9. Transmission Electron Microscope

Fresh liver tissues were sampled at a volume of 1 mm × 1 mm × 2 mm and were fixed in fixative (2.5% glutaraldehyde in pH 7.4 cacodylate buffer) for 2 h at room temperature. The liver tissues were washed three times in PBS and post-fixed in 1% osmium tetroxide. The samples were dehydrated through ascending concentrations of alcohol and post-embedded in Araldite. Ultra-thin sections were stained with uranyl acetate and lead citrate, and images were captured with a HITACHI HT7700 transmission electron microscope (Tokyo, Japan).

### 2.10. Mitochondrial Membrane Potential (MMP)

Fresh liver tissues (100 mg each) were obtained from fish in all CP groups following deep anesthesia and then cut into pieces using scissors before mitochondrial isolation using a commercial kit (Beyotime, Catalog No. C3606, Beyotime Biotechnology, Shanghai, China). The MMP assay was conducted using the JC-1 fluorescent probe method (Beyotime, Catalog No. C2006) for rapid visualization in the dark using green and red fluorescence (Ex = 490 nm and Em = 530 nm for JC-1 monomers; Ex = 525 nm and Em = 590 nm for J-aggregates).

### 2.11. Reactive Oxygen Species (ROS) Detection

The fish were anesthetized with 0.01% MS-222. The liver tissue was quickly removed and cleaned with PBS for 3 times. Liver tissue samples were dissected into 1 cm × 1 cm × 1 cm pieces using surgical scissors, digested with 0.25% trypsin, and incubated on an orbital shaker for 10 min. Single-cell suspension was prepared, the fluorescent probe was added, and the precipitation was suspended with diluted DCFH-DA (Beyotime Biotechnology, Shanghai, China). Cell precipitation was incubated at 27 °C for 40–60 min, centrifuged at 7000× *g* for 10 min, and the supernatant was collected and deposited with PBS. Photographs of the precipitate were taken using a confocal fluorescence microscope (Eclipse-Ti-S, Niko, Tokyo, Japan) after it had been suspended and mixed. The absorbance was detected at 525 nm by a fluorescence microplate to calculate the ROS content. Measurement of fluorescence signal intensity of larvae by Image J software (Version 1.53k, National Institutes of Health, Bethesda, MD, USA).

### 2.12. Real-Time Quantitative PCR

Total RNA was extracted from the liver using Trizol reagent (Invitrogen, Carlsbad, CA, USA) following standard procedures and quantified using a NanoDrop 2000 (Thermo, Waltham, MA, USA). cDNA was synthesized from 2 μL of total RNA using the PrimeScript^®^ Reverse Transcription Kit and gDNA Eraser (TaKaRa, Dalian, China). Real-time PCR was performed in the CFX96 Real-Time PCR Detection System (Bio-Rad, Hercules, CA, USA). The sequences of all primers used in this study are listed in Table 2. The Real-time PCR reaction (10 μL) included 5 μL of ChamQ Universal SYBR qPCR Master Mix (Vazyme, Nanjing, China), 3 μL of ddH_2_O, 0.5 μL of each primer, and 1 μL of cDNA. The qRT-PCR protocol in this study was conducted as follows: 95 °C for 2 min, followed by 40 cycles of denaturation 95 °C for 5 s, annealing at the melting temperature (Tm) for 30 s, then 95 °C for 5 s, extension at 65 °C for 5 s, and a final extension at 95 °C for 15 s. The qRT-PCR was performed using a CFX96 Real-Time PCR Detection System (Bio-Rad, Hercules, CA, USA). β-actin and 18S rRNA were chosen as internal reference genes for normalization. The results were calculated using the 2−^ΔΔCT^ method.

### 2.13. Western Blotting

Proteins were separated from cells using a Cold Lysis Buffer (Beyotime, Shanghai, China) containing a mixture of protease and phosphatase inhibitors. Protein content in the cell supernatant was measured using a BCA protein quantification kit (Beyotime, Shanghai, China). Samples containing equal amounts of protein (20 μg) were separated by 10% SDS-polyacrylamide gel electrophoresis and then transferred to polyvinyl alcohol difluoride membranes (Bio-Rad Co., USA). Membranes were blocked and incubated with primary antibodies overnight at 4 °C (SIRT1, 1:2000, ZenBio (Chengdu, China), R25721; Yap, 1:1000, ABclonal (Wuhan, China), A1001; phospho-Yap (p-Yap), 1:1500, ZenBio, 381297; Trfc, 1:2000, ABclonal, A21622; Trf, 1:2000, ABclonal, A1448; Gpx4, 1:1500, ZenBio, 381958; β-actin, 1:3000, ZenBio, 380624; Lamin B, 1:1000, Abclonal, A1910). After washing four times with TBST (5 min each), proteins were visualized by incubating the membranes with the corresponding HRP-labeled secondary antibodies for 2 h at 25 °C. The membranes were washed four times for 5 min each with TBST using the ECL chemiluminescence kit. Finally, protein density analysis was performed using the Gel-Pro Analyzer (Rockville, MD, USA). All proteins were normalized for relative expression to β-actin or LaminB1 (Appendix A).

### 2.14. Immunohistochemistry

The paraffin-embedded liver sections were prepared for immunohistochemical analysis. De-paraffinization was performed with ethanol, while xylene was used for hydration. Samples were incubated with three percent hydrogen peroxide to block endogenous peroxidases. Primary antibody (Gpx4, 1:1000; ZenBio) incubation occurred overnight at 4 °C, followed by washing with PBS every 5 min. PBS washing was followed by continuous incubation with secondary antibody (1:5000; Affinity, Nanjing, China) and color development with DAB. A neutral resin adhesive sealed the nuclei, which were then dehydrated, cleared, and counterstained with hematoxylin. Stained sections were examined with an Olympus microscope (Olympus corporation, Tokyo, Japan).

### 2.15. Immunofluorescence Staining

Paraffin-embedded liver sections 5 μm thick were fixed and washed twice with PBS. Sections were blocked with bovine serum albumin and then incubated with anti-TRFC and YAP antibodies. Subsequently, the sections were incubated with FITC- or Cy3-coupled secondary antibodies for 2 h at room temperature (Ex = 525 nm and Em = 488 nm for FITC; Ex = 570 nm and Em = 550 nm for Cy3). Cell nuclei were counterstained with DAPI. Images of the cells were captured using a fluorescent microscope (IX70, Olympus, Tokyo, Japan).

### 2.16. Statistical Analysis

Data normality and homogeneity were assessed using D’Agostino–Pearson and Bartlett tests, respectively. If the data met these criteria, a one-way ANOVA with Duncan’s multiple comparison test was performed to compare main factors. The data were analyzed using the PROC MIXED procedure of SAS 9.3 (SAS Inst. Inc., Cary, NC, USA) with the following model:*Y_ijkl_* = *μ* + *T_i_* + *P_i_* + *S_k_* + *C(_K_)_l_* + *T* × *S_ik_* + *e_ijkl_*,
where *Y_ijkl_* refers to the dependent variable; *μ*, the overall mean; *T_i_*, the fixed treatment effect; *P_j_*, the random period effect; *S_k_*, the random square effect; *C(_k_)_l_*, the random effect of the lth steer in the kth square; *T × S_ik_*, the interaction between the *i*th treatment and the *k*th square; and *eijkl*, the error residual. The linear and quadratic effects of increasing CP levels were analyzed using the CONTRAST procedure in SAS 9.3 (SAS Institute Inc., Cary, NC, USA). The Kenward–Roger option was used to calculate the degrees of freedom. Differences among the means of different treatments were tested using Duncan’s test. Effects were considered significant at *p* < 0.05. All values are presented as least squares means ± standard errors (SEs).

## 3. Results

### 3.1. Growth Performance and Whole-Body Composition

Table 3 present the effects of CP levels on the growth performance of *M. salmoides* observed in this study. Compared to the FM group, the CP36 group exhibited notable reductions in FBW (*p* < 0.001), PWG (*p* < 0.001), SGR (*p* < 0.001), HSI (*p* < 0.001), VSI (*p* < 0.001), and ISI (*p* = 0.001). FI (*p* = 0.119) and FE (*p* = 0.061) were lower in the CP48 and CP60 groups compared to the FM group. Whole-body composition, including crude protein (*p* = 0.432), crude lipid (*p* = 0.448), and ash content (*p* = 0.446), showed no significant differences among the groups (Table 4). However, the CP24 group had significantly lower moisture content compared to the other groups (*p* = 0.039). No significant effects of dietary crude protein level were observed for protein production value (PPV) (*p* = 0.340) and lipid production value (LPV) (*p* = 0.273). The broken-line regression model, using SGR as the response variable, identified 23% as the optimum replacement ratio of FM by CP in the feed formulation (Figure 1).

### 3.2. Serum and Liver Biochemical Parameters

Table 5 summarizes the serum and liver biochemical parameters. Serum TG and TC levels gradually increased in the CP24 and CP36 groups (*p* < 0.001). The CP48 group exhibited significantly higher HYP levels than the other groups (*p* < 0.001). Serum AKP activities initially increased and then decreased as CP replacement levels rose (*p* < 0.001). AKP activities peaked in the CP36 group compared to the FM group (*p* < 0.001). Liver ALT and AST activities were significantly higher in the CP60 group than in all other groups (*p* < 0.001).

### 3.3. Liver Morphology and Inflammation-Related Genes

Histological examination of liver morphology (Figure 2) demonstrated that the FM group exhibited bright red coloration and smooth hepatic architecture, with intact, well-organized lobules and centrally located, round hepatocyte nuclei. In contrast, the CP36, CP48, and CP60 groups displayed yellowish, blunted livers. The CP36 and CP48 groups showed inflammatory cell infiltration and vascular wall thickening. Hepatocyte outlines were indistinct in the CP48 and CP60 groups, with nuclei dissolving and disappearing. Masson staining indicated collagen fiber deposition in the CP36, CP48, and CP60 groups. In each CP group, there was an obvious increase in IL-1β and TNF-α expression as compared to the control group (*p* < 0.05). After feeding CP60, there was a downregulation of the anti-inflammatory cytokine IL-10 and TGF-β2 (*p* < 0.05, Figure 3).

### 3.4. Oxidation Indicators and Fe^2+^ Content

Mitochondrial function was assessed using transmission electron microscopy (TEM), JC-1 membrane potential staining, and ATP content measurement. Most liver mitochondria in the FM and CP24 groups appeared normal, with oval shapes, intact membranes, and well-distributed cristae (Figure 4A). However, in the CP60 group, the outer mitochondrial membrane was ruptured, and mitochondrial cristae were absent, indicating characteristics of ferroptotic mitochondria. Correspondingly, liver MMP and ATP content decreased in the CP60 group compared to the FM and CP24 groups (*p* < 0.05) (Figure 4B,C).

ROS accumulation, a marker of ferroptosis, was significantly higher in the CP60 group compared to FM and CP24 (*p* < 0.05) (Figure 4D,E). Furthermore, the levels of the antioxidant factor GSH and GST activities in liver tissues were reduced with higher CP substitution levels (Figure 4F,G). Liver GSH levels in the CP48 group exhibited the lowest GSH levels (*p* < 0.05), while GST activity was lowest in the CP36 group (*p* < 0.05) (Figure 4H). MDA levels were highest in the CP48 group and the CP24 group also showed higher levels than the FM group (*p* > 0.05). MDA levels progressively increased in the CP36 group and reached a peak in the CP48 group (*p* < 0.05).

Iron overload, a hallmark of ferroptosis, was assessed via liver Fe^2+^ content in the different CP replacement FM groups using an iron assay kit. The results showed that the Fe^2+^ content of CP48 and CP60 groups was significantly increased compared with other groups (Figure 4I).

### 3.5. Hepatic Ferroptosis-Related Genes

CP levels altered the expression of ferroptosis-related genes (Figure 5). The mRNA expression of *Trf* increased in the CP12, CP24, and CP36 groups but decreased in the CP48 and CP60 groups. The *Trf* mRNA expression did not significantly differ between the FM, CP12, CP24, and CP36 groups, whereas levels were significantly reduced in CP48 and CP60 (*p* < 0.05). *Lpcat3* mRNA expression remained unaffected in the CP12, CP24, and CP36 groups (*p* > 0.05) but was higher in the CP24 and CP36 groups (*p* < 0.05) (Figure 5A–C). *Acsl4* and *Ptgs2* mRNA expression levels increased with increasing CP levels (*p* < 0.05) (Figure 5D,E). Similar to *Trf* mRNA expression, TRF protein levels did not significantly differ among the CP12, CP24, CP36, and CP48 groups but showed significant differences in the CP60 group compared to the FM group (*p* < 0.05) (Figure 5F,H). Notably, the CP48 and CP60 groups exhibited higher TRF protein levels (*p* < 0.05).

### 3.6. SIRT1-YAP-TRFC Pathway

The expression levels of *Sirt1* and *Yap* mRNA decreased as CP replacement increased. *Sirt1* mRNA expression levels were significantly lower in the CP48 and CP60 groups compared to the control FM group (*p* < 0.05) (Figure 6A). In the CP36 group, the expression level of *Yap* mRNA began to decrease, and it was significantly lower in the CP60 group than in the control group (*p* < 0.05) (Figure 6B). *Trfc* mRNA expression levels increased with the increase in CP substitution levels, and in the CP60 group, the expression level was significantly higher than that in the FM, CP12, CP24, and CP36 groups (*p* < 0.05) (Figure 6C). The SIRT1 protein level correlated with mRNA expression and decreased to the lowest level in the CP48 and CP60 groups (*p* < 0.05) (Figure 6D,E). YAP protein level gradually decreased with the increase in CP replacement rate, although no significant differences were observed among the CP36, CP48, and CP60 groups (*p* > 0.05). However, these levels were higher than those in the other three groups (*p* < 0.05) (Figure 6D,F). Nuclear TRFC and YAP protein levels increased with higher CP substitution, with the CP48 and CP60 groups showing significantly higher levels than the FM group (*p* < 0.05) (Figure 6D,G,H). IHC of GPX4 protein in the liver of juvenile *M. salmoides* revealed a gradual decrease in the brownish-yellow positive signal compared to the FM group (Figure 6I). This indicated reduced GPX4 protein accumulation in the cytoplasm with the increase in CP content. The subcellular localization of YAP and TRFC, assessed via immunofluorescence assays (Figure 7), demonstrated increased TRFC expression and nuclear enrichment of YAP protein in hepatocytes in the CP48 and CP60 groups. YAP, a transcriptional cofactor, was shown to translocate from the cytoplasm to the nucleus under these conditions.

### 3.7. Gossypol and SIRT1 with Molecular Docking

Subsequently, we investigated the mechanism of SIRT1 inhibition by gossypol using molecular docking (Figure 8A,B). The results revealed that the gossypol small molecule fits into the ligand-binding domain pocket of SIRT1, a key regulatory domain for its activity, and forms three hydrogen bonds at ILE 270, Val 266, and ARG 274 rather than non-specific bonds.

## 4. Discussion

### 4.1. Effects of Partial Substitution of FM with CP on Growth Performance Indices of M. salmoides

Studies have indicated that incorporating moderate amounts of vegetable protein into aquatic animal feeds can promote growth, while excessive levels may hinder it by delaying development [25]. The results of this study demonstrated that the final weight, weight gain rate, and specific growth rate of juvenile *M. salmoides* remained unaffected when CP substitution in the diet was below 24%. However, the growth performance of juvenile *M. salmoides* declined when the substitution rate exceeded 24%, with the lowest FBW, PWG, and SGR observed at a 60% substitution rate. Previous studies have explored the substitution of CP for FM in aquatic animal diets. For example, it was observed that replacing 30% of FM with dephenolized CP had no effect on the growth performance of spot prawns, but higher substitution rates significantly reduced their growth [11]. Likewise, large yellow croaker experienced no growth impact with a 9% FM replacement by dephenolized CP [8]. Black seabream exhibited no changes in PWG or SGR when 16% of FM was replaced by fermented CP [25]. However, excessive CP in diets can introduce toxins such as gossypol, which affect growth by inhibiting development. Of the FM replaced with 60% CP decreased FI and FE. The results indicated that CP could reduce the feed efficiency of *M. salmoides*. The reduced growth performance may be partially attributed to decreased feed intake caused by lower palatability. Free gossypol readily binds with lysine, reducing its activity and making lysine the first limiting amino acid in feed [26,27]. A deficiency in lysine not only diminishes the palatability of the feed but also lowers the efficiency of protein conversion [28]. Adding lysine to plant-based feed can not only replace the high cost of FM but also help reduce phosphorus emissions to the environment, thereby minimizing pollution, as plant protein contains less phosphorus than FM. In addition, factors such as cottonseed detoxification processing technology, feeding methods, growth stages of experimental subjects, and feed palatability also influence FM substitution efficiency [6]. Improving detoxification methods, eliminating anti-nutritional factors, and supplementing essential amino acids are recommended strategies to optimize plant protein utilization.

### 4.2. Effects of CP Replacement of FM on the Nutrient Deposition of M. salmoides

Protein is the primary energy source for growth and metabolism in fish, especially carnivorous species, which have high dietary protein requirements. The quality of dietary protein is assessed by its absorption and changes in body composition. This study showed no significant effect of CP substitution for FM on crude fat or crude protein content in juvenile *M. salmoides*. However, the moisture content was lowest when FM was replaced with 24% CP in the diet and increased when CP substitution reached 48%. Higher moisture content may result from reduced feed intake due to poor palatability. Similar findings were reported in studies with snapper (*Pagrus major*) [29]. Zhao et al. (2021) observed no significant changes in moisture, ash, crude protein, or crude fat content in anchovy when less than 50% of fishmeal (FM) was replaced with cottonseed protein concentrate [7]. Anderson et al. (2016) reported that replacing 75% of FM with cottonseed meal did not alter moisture, ash, crude protein, or crude fat levels in juvenile black sea bass [30]. In contrast, Alam et al. (2018) found that substituting 75% and 100% of FM with cotton meal significantly increased body fat content in southern flounder (*Paralichthys lethostigma*) [9]. These differences highlight how dietary CP influences nutrient deposition, with variations depending on feed ingredients, substitution levels, species adaptability, and study duration [30]. These results suggest that dietary CP not differ significantly from nutrient deposition.

### 4.3. Effects of Substituting FM with CP on Hepato-Intestinal Development of M. salmoides

The intestinal tract and liver are essential digestive and immune organs in fish. Their normal development serves as a key indicator of fish health and can directly reflect feed utilization efficiency. In this study, the VSI, HSI, and ISI declined as the proportion of FM replaced by CP in the diet increased. All three ratios significantly decreased when the replacement rate exceeded 24%. Previous research has demonstrated that substituting FM with high levels of plant protein can trigger intestinal inflammatory response, disrupting the structure of intestinal villi and impairing digestion and absorption [31]. Similarly, our study observed that high-level CP substitution (48–60% FM replacement) suppressed growth performance in *M. salmoides*. Sun’s findings indicated that as the proportion of fermented CP replacing FM in the diet increased [25], the HSI and VSI of young black sea bass significantly decreased [32]. The reduction in the HSI may be attributed to an uneven distribution of amino acids in the feed, resulting in an imbalance in amino acid catabolism during lipid synthesis [32]. Thus, excessive substitution of CP sources for FM beyond a certain percentage can compromise fish nutritional status and hinder feed utilization.

### 4.4. Effects of CP Replacement of FM on Liver Morphology, Biochemical Indices, and Inflammation-Related Genes of M. salmoides

This study revealed that when the proportion of CP replacing FM exceeded 24%, the liver of juvenile *M. salmoides* gradually exhibited a lighter, more yellowish, and blunter appearance. Histological observations revealed hepatocyte enlargement, blurred outlines, disordered hepatic lobule arrangement, infiltration of inflammatory cells, and local necrosis at a 60% replacement ratio. Furthermore, collagen deposition occurred in the liver tissues of the high CP replacement FM group, indicating potential liver fibrosis development. Chronic liver injury of this nature is associated with increased collagen-dominated extracellular matrix synthesis and the activation of hepatic stellate cells into myofibroblasts and fibroblasts. Serum and liver homogenate-based indices effectively reflect changes in liver function. AST primarily transfers the amino group from aspartic acid to α-ketoglutaric acid, forming oxaloacetic acid and glutamic acid. ALT transfers the amino group from alanine to α-ketoglutarate, producing pyruvate and glutamate [33]. Oxaloacetate is crucial in the tricarboxylic acid cycle, providing essential energy for the body [34]. Elevated activities of AST and ALT are indicative of liver damage [35], while AKP serves as a biomarker for biliary stasis and liver fibrosis [36]. In the present experiment, results indicated that when the proportion of CP replacing FM exceeded 24%, the serum TG and TC levels significantly increased. This suggests that the liver lipid content of the fish was increased. Serum AKP activity was highest at the replacement ratio of 36%, indicating potential bile retention in the liver. Transaminase, including ALT and AST, is indicative of liver damage when present in elevated serum levels. Liver AST activity was highest when the replacement ratio was 60%, and ALT activity peaked at the 48% replacement ratio. These findings indicated that excessive substitution of FM with CP can lead to substantial liver damage in fish, with the degree of damage varying according to fish species’ tolerance to plant protein products. The research by He et al. revealed that substituting 18% of FM with CP concentrate has been reported to result in liver tissue yellowing and diffuse lipid vacuolation in some hepatocytes of *M. salmoides* [14]. This study not only confirms the negative effects of CP on growth and liver function as reported by He et al., but more importantly elucidates its mechanism of action from perspectives of ferroptosis, signaling pathways, and ultrastructural changes, thereby providing a more comprehensive theoretical basis for the safe utilization of CP in aquatic feeds. Wang et al. found that replacing CP with 27% FM significantly increased liver ALT and AST activity and led to hepatocyte atrophy in juvenile common carp [37]. Collectively, these results suggest that excessive CP substitution for FM induces liver damage, with severity varying depending on the fish species and processing techniques of plant protein products. In fish, cytokines are broadly categorized into pro-inflammatory factors (including IL-1β and TNF-α) and anti-inflammatory factors (e.g., IL-10, TGF-β1), both playing crucial roles in mediating immune responses. Studies have found that CP reduces anti-inflammatory factors and promotes the expression of pro-inflammatory factors, thereby promoting inflammation in *M. salmoides* [38]. In this study, we found that high-level CP could enhance the relative expression of pro-inflammatory factors (IL-1β and TNF-α) and decrease the relative expression of anti-inflammatory factors (IL-10 and TGF-β2) of juvenile *M. salmoides*, agreeing with previous studies, which suggested that high-level CP could decrease the immunity and trigger the inflammation of juvenile *M. salmoides*.

### 4.5. Excess Substitution of CP for FM Caused Hepatic Fibrosis and Was Associated with Ferroptosis

Liver fibrosis is pathologically defined by excessive deposition of extracellular matrix (ECM) proteins, particularly collagen subtypes, in chronic hepatic injury contexts [39]. With Masson’s trichrome staining, a reliable method for highlighting collagen disposition, we found that high-level CP effectively increases collagen fiber accumulation. Hydroxyproline (HYP), a major component of collagen, elevates during liver fibrosis [40,41]. Notably, high-level CP successfully increases serum HYP content. These collective results highlight high-level CP induced hepatic fibrosis of *M. salmoides*. To further investigate the effects of excessive CP substitution for FM on the liver fibrosis by ferroptosis mediated in juvenile *M. salmoides*. This study used TEM to examine the ultrastructure of hepatocytes. The results demonstrated that when the proportion of CP replacing FM was 60%, there was a significant change in mitochondrial morphology, characterized by mitochondrial atrophy and loss of cristae compared to the FM group. This is a typical morphological feature of cell ferroptosis. In mice, ferroptosis has been linked to liver damage induced by acetaminophen (APAP), lipopolysaccharide (LPS) combined with D-galactose amino, and microplastics (MPs) [42,43,44]. MMP is a prerequisite for maintaining oxidative phosphorylation of mitochondria to produce ATP, and the stability of MMP is conducive to maintaining the normal physiological function of cells [45,46]. MMP was upregulated in the CP24 group, indicating that it could improve mitochondrial function. However, MMP was downregulated in the CP60 group, indicating that it would lead to abnormal mitochondrial membrane potential and negatively affect mitochondrial function. Ferroptosis is an oxidative stress-related programmed cell death mechanism characterized by iron-dependent lipid peroxide accumulation. This process involves elevated iron ion concentrations, increased ROS, and lipid peroxidation [47]. GPX4 plays a crucial role in mitigating ferroptosis by eliminating lipid peroxides.

In this study, the liver of juvenile *M. salmoides* was examined for Fe^2+^, ROS, lipid peroxidation product MDA content, and antioxidant factor GSH content in control and test groups. GSH serves as a cofactor of GPX4 and inhibits ferroptosis. The results revealed that in the CP60 group, the accumulation of ROS in the liver tissues of *M. salmoides* was higher than that in the FM and CP24 groups. In the CP48 group, the level of GSH in the liver were significantly reduced. Liver tissue MDA content gradually increased in the CP36 group and reached a maximum in the CP48 group. The study also examined ferroptosis-related gene expression levels and found that hepatic TRF protein expression levels were highest in the CP60 group compared to the FM group. GPX4 mRNA and protein levels were significantly lower in the CP48 and CP60 groups. *Acsl4* and *Ptgs2* mRNA levels were significantly higher in the CP36, CP48, and CP60 groups compared to the control FM group. *Lcpat3* mRNA expression was the highest in the CP48 and CP60 groups.

Excessive substitution of FM with plant protein has been studied to result in elevated levels of oxidative stress ROS in the intestine of *Schizothorax prenanti*, causing damage to the intestinal barrier [48]. Additionally, replacing FM with dephenolized CP has been shown to reduce the antioxidant immune function of the liver in silver perch, leading to elevated ROS levels in the liver [49]. Circulating Fe^3+^ binds to and is transported by TRF. Once it enters the cell via TFRC, it undergoes reduction and is released into the cytoplasmic unstable iron pool (LIP) and the excess iron is stored in ferritin. Iron has the capability to react directly with polyunsaturated fatty acids found in cell membranes and other biological membranes due to its instability and high reactivity as Fe^2+^. This reaction generates substantial amounts of lipid peroxides through the fenton reaction in conjunction with hydroxyl radicals like H_2_O_2_ [50]. Consequently, iron overload can induce ferroptosis and exacerbate liver injury. The over-substitution of CP for FM decreases hepatic antioxidant GSH levels and increases ROS accumulation in hepatocytes. ROS, in turn, initiates lipid peroxidation in polyunsaturated fatty acid-rich lipid bilayers, such as cell and mitochondrial membranes. Additionally, excess Fe²^+^ directly catalyzes lipid peroxidation, leading to the production of more lipid peroxides. This destructive feedback loop culminates in ferroptosis and liver damage.

### 4.6. Replacement of FM by Excess CP Caused Ferroptosis Through SIRT1-AP-TRFC Signaling Pathway

Liver damage in *M. salmoides* due to excessive FM replacement with CP was closely associated with ferroptosis, as evidenced by hepatic iron overload, mitochondrial morphological alterations, and lipid peroxide accumulation. To investigate potential regulatory signaling pathways, this study considered the involvement of the SIRT1-YAP-TRFC pathway. Previous studies have shown that CP hydrolysates replacing FM can impact hepatic *Sirt1* mRNA expression, affecting the amino acid metabolism of pomfrets. Sirt1 is a prominent member of histone deacetylases, widely involved in oxidative stress, inflammation, cell metabolism, Alzheimer’s disease, and cardiovascular diseases [17,51,52]. For instance, SIRT1-mediated deacetylation of transcription factor C/EBP-β inhibits YAP, reducing hypertrophic scarring [53]. Overexpression of Sirt1 can also inhibit the Hippo/YAP signal and reduce periarterial sclerosis [19]. Recent studies have suggested that SIRT1 may regulate downstream transcriptional coactivators, participating in ferroptosis-induced liver injury. Ubiquitin-specific peptidase 22 (USP22) stabilizes SIRT1 protein, reduces P53 acetylation, and upregulates SLC7A11, ultimately inhibiting ferroptosis and myocardial cell damage caused by ischemia-reperfusion [54]. Furthermore, the Merlin–Hippo pathway has been implicated in ferroptosis regulation [55]. Activated YAP promotes ferroptosis in epithelial cancer cells by upregulating ferroptosis-regulatory genes such as ACSL4 and TRFC [21]. O-GlcNAc glycosylation modifies the YAP/TFRC pathway, increasing susceptibility to ferroptosis induced by GSH peroxidase 4 inhibitors (RSL4) [56]. Based on this, we hypothesize that excessive substitution of CP for FM contributes to liver injury associated with ferroptosis by influencing the SIRT1-YAP-TRFC pathway. Our results demonstrated that as the proportion of CP replacing FM increased, Sirt1 and Yap mRNA levels decreased, while Trfc mRNA levels rose. Phosphorylated YAP (p-YAP) protein levels were significantly higher in the CP60 group compared to other groups. SIRT1 protein levels followed a similar pattern to their mRNA levels, reaching their lowest levels in the CP48 and CP60 groups. Additionally, p-YAP protein levels decreased progressively with increasing CP replacement rates, indicating substantial translocation of YAP into the nucleus. YAP protein expression levels in both the nucleus and TRFC increased with higher CP replacement rates, with the CP48 and CP60 groups exhibiting the highest expression of these proteins. This study also observed that SIRT1 deacetylation promotes the nuclear accumulation of YAP, which directly regulates the TRFC promoter. This regulation enhances TRFC protein expression, leading to iron imbalance and contributing to ferroptosis in the liver. Although further research is needed, we hypothesize that the accumulation of gossypol—resulting from CP substitution for FM—may influence SIRT1 expression, thereby contributing to the occurrence of liver ferroptosis. Notably, this study is the first to report on the specific mechanism by which CP substitution regulates SIRT1 expression.

## 5. Conclusions

In a word, our findings showed that high-level CP replacing FM may suppress growth performance. Meanwhile, high-level CP significantly upregulated the relative expression of pro-inflammatory factors (IL-1β and TNF-α), while suppressing anti-inflammatory mediators (IL-10 and TGF-β2). Higher levels of CP substitution may induce liver ferroptosis by activating the SIRT1-YAP-TRFC pathway, leading to liver damage in *M. salmoides*. Without compromising the growth performance of juvenile *M salmoides*, broken-line regression analysis using SGR as the indicator determined that the optimal replacement level of FM with CP in the diet was 23%.

## Figures and Tables

**Figure 1 biology-14-00748-f001:**
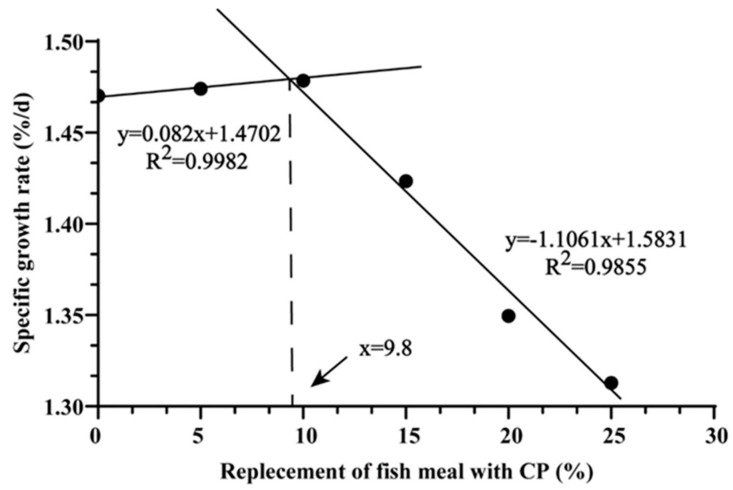
The relationship between SGR and dietary CP replacement level in *M. salmoides*.

**Figure 2 biology-14-00748-f002:**
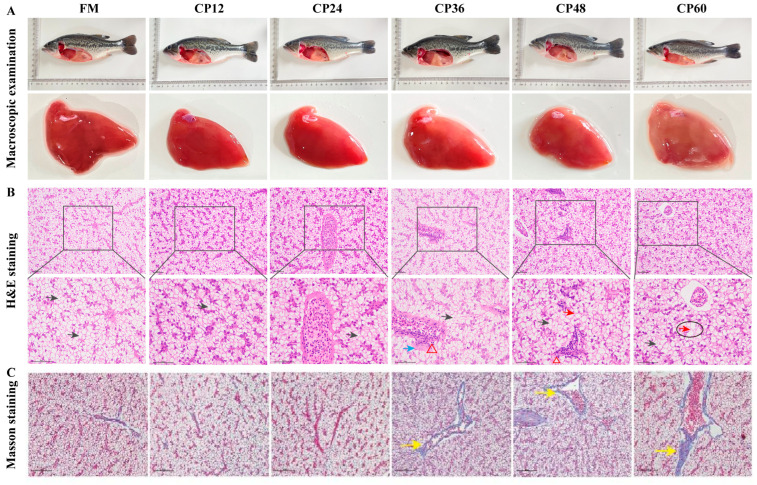
Histology of liver in juvenile *M. salmoides* fed diets containing different levels of CP.: (**A**) Fish body and liver images; (**B**) Hematoxylin and eosin (H&E); (**C**) Masson’s trichrome staining. H&E: 100 μm and 20 μm, Masson: 20 μm. Black arrow: hepatocytes; Blue arrow: vascular wall hyperplasia; Red arrow: hepatic nucleolysis; Red triangle: inflammatory cell infiltration; Black circle: local necrosis of hepatocytes; Yellow arrow: collagen fiber deposition.

**Figure 3 biology-14-00748-f003:**
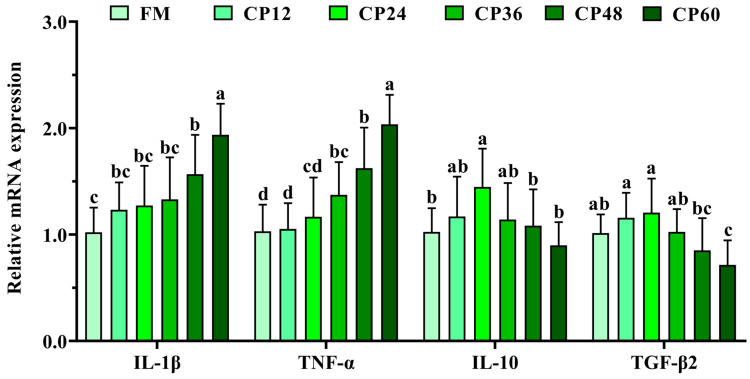
Effects of CP on the expression of pro-inflammatory cytokines (IL-1β and TNF-α), anti-inflammatory cytokines (IL-10 and TGF-β2) in the liver of *M. salmoides*. All data represent means ± SEM of three replicates. Bars with different letters indicate significant differences by Duncan’s test (*p* < 0.05, *n* = 9).

**Figure 4 biology-14-00748-f004:**
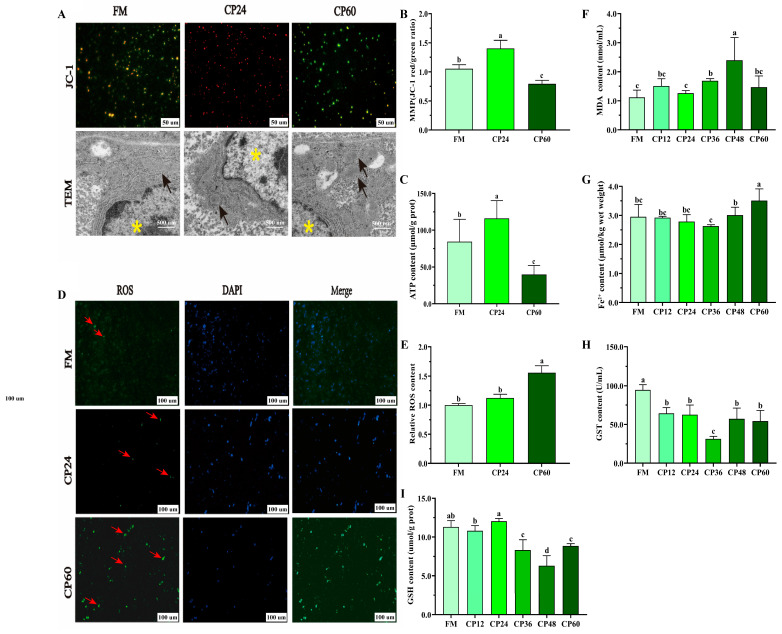
Excess CP caused liver damage and was associated with ferroptosis in the *M. salmoides*. (**A**): Liver mitochondria were collected for JC-1 staining (Scale: 50 um) and Transmission Electron Microscope (TEM) examination (Scale: 500 nm, *n* = 9); black arrows: mitochondrion; yellow asterisk: hepatic nucleus; (**B**): The ratio of red and green fluorescence of JC-1; (**C**): ATP contents; (**D**): fluorescence probe reactive oxygen species (ROS) staining, red arrows represent ROS (*n* = 9); (**E**): relative ROS fluorescence value statistics; (**F**–**H**): glutathione (GSH), glutathione S-transferase (GST), and malondialdehyde (MDA), respectively; (**I**): Liver Fe^2+^ level. All data represent means ± SEM of three replicates. Bars with different letters indicate significant differences by Duncan’s test (*p* < 0.05, *n* = 12).

**Figure 5 biology-14-00748-f005:**
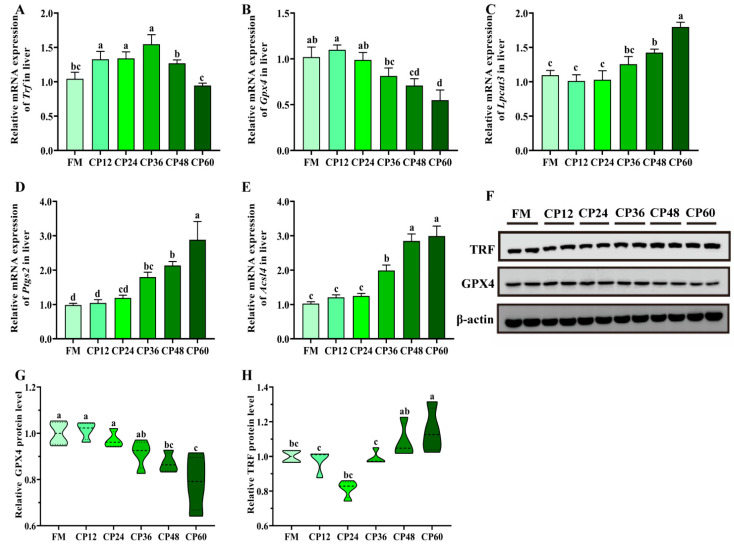
Effect of excess cotton protein (CP) on ferroptosis-related genes in the liver of juvenile *M. salmoides*. (**A**–**E**): Effect of dietary CP on mRNA expression of TRF, GPX4, LPCAT3, PTGS2, and ACSL4 in the liver of juvenile *M. salmoides*; (**F**): Representative western blot of TRF, and GPX4 in the liver; (**G**,**H**): Protein levels of GPX4 and TRF. Data represent means ± SEM of three replicates. Bars with different letters indicate significant differences by Duncan’s test (*p* < 0.05, *n* = 12).

**Figure 6 biology-14-00748-f006:**
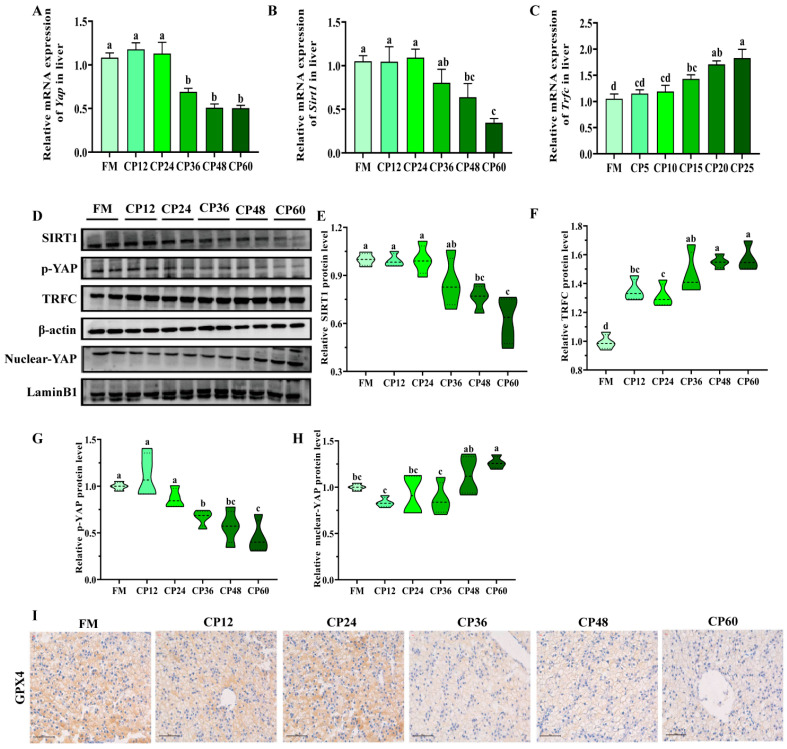
Effect of dietary CP on *SIRT1-YAP-TRFC* pathway in the liver of juvenile *M. salmoides*. (**A**–**C**): mRNA expression of SIRT1, YAP, and TRFC. (**D**): Representative Western blot images of nuclear YAP, SIRT1, p-YAP, and TRFC; Protein levels of (**E**) SIRT1; (**F**) p-YAP (Ser 127); (**G**) TRFC; (**H**) nuclear YAP; (**I**): liver sections immunohistochemically stained for GPX4 (Scale = 50 μm), the brown signal represents positive expression. Data represent means ± SEM of three replicates. Bars with different letters indicate significant differences by Duncan’s test (*p* < 0.05, *n* = 12).

**Figure 7 biology-14-00748-f007:**
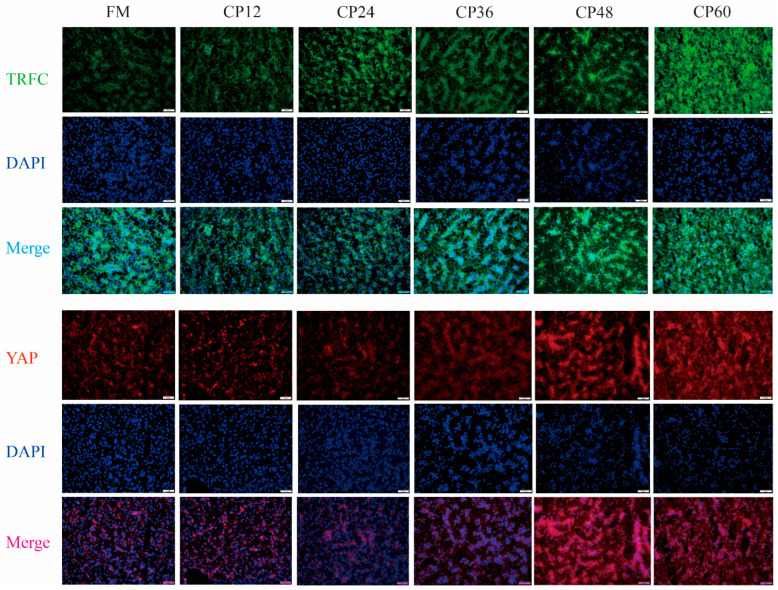
Fluorescence microscopy images of DAPI (Blue), TRFC (Green), and YAP (Red) in the liver of *M. salmoides* fed with different six studied diets. Replacement of fish meal with CP increased hepatocyte expression of TRFC (Green), and nuclear enrichment of YAP protein in the presence of CP48 and CP60 (Scale = 20 μm, *n* = 9).

**Figure 8 biology-14-00748-f008:**
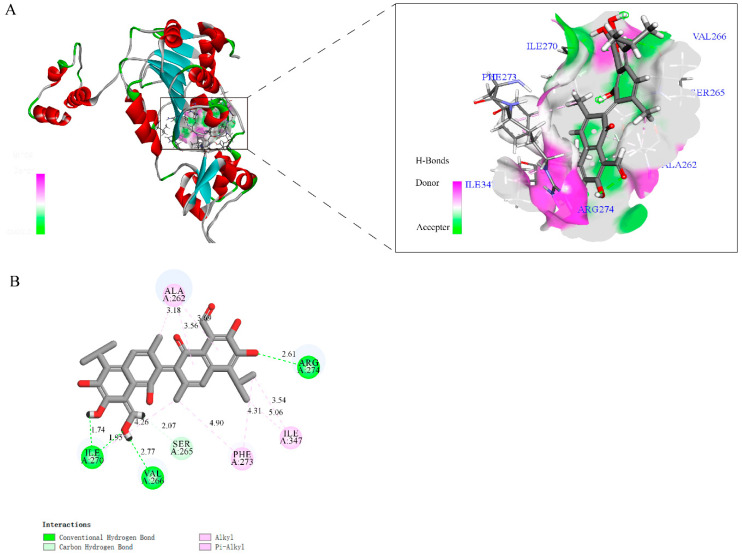
The model of the mechanism of excess CP mediated by ferroptosis following liver damage in M. salmoides. (**A**): The binding mode between micromolecule gossypol and SIRT1 protein, showing interactions between gossypol and key residues. (**B**): A two-dimensional interaction map of gossypol and SIRT1. Excess CP reduced SIRT1 expression to promote the nuclear accumulation of YAP, YAP can directly regulate the TRFC promoter, and promote the expression of TRFC protein, which upregulated the levels of ferroptosis. Cellular ROS content was increased by excess CP-induced ferroptosis, ultimately leading to ferroptotic cell death.

**Table 1 biology-14-00748-t001:** Composition and nutrients content of basal diet (g/kg).

Ingredients	FM	CP12	CP24	CP36	CP48	CP60
Soya bean oil ^1^	40.2	44.0	47.8	51.6	55.4	59.2
Bentonite	84.8	69.0	52.6	34.4	17.2	0.0
Fish meal	520.0	458.8	397.6	336.4	275.2	214.0
Chicken meal	160.0	160.0	160.0	160.0	160.0	160.0
Wheat gluten	70.0	70.0	70.0	70.0	70.0	70.0
Cottonseed protein	0.0	70.0	140.0	210.0	280.0	350.0
Wheat flour	50.0	50.0	50.0	50.0	50.0	50.0
Tapioca	34.0	34.0	34.0	34.0	34.0	34.0
Lysine	0.0	0.5	1.6	4.3	6.2	8.2
DL-methionine	2.0	2.7	3.4	4.3	5.0	5.6
CaH_2_PO_4_	10.0	12.0	14.0	16.0	18.0	20.0
Choline chloride	4.0	4.0	4.0	4.0	4.0	4.0
Mineral element premix ^2^	10.0	10.0	10.0	10.0	10.0	10.0
Vitamin premix ^3^	15.0	15.0	15.0	15.0	15.0	15.0
Total	1000	1000	1000	1000	1000	1000
Nutrients content ^4^						
Crude protein	518.5	521.2	521.0	522.3	521.2	520.3
Crude lipid	78.8	80.8	81.2	80.3	80.7	79.4
Moisture	49.8	48.8	49.7	50.0	50.2	48.5

^1^ Soya bean oil (Dongguan Yihai Jiali Cerui Starch Co., Ltd. Dongwan, China), Fish meal (contains 68% protein and 13.75% lipid) (Tianbao Grain and Feed Trading Co., Ltd. Lima, Peru), Chicken meal (Sichuan Xinrui Feed Technology Co., Ltd. Meishan, China), Wheat gluten (Qionglai Sonaker Biotechnology Co., Ltd. Qionglai, China), Cottonseed protein (contains 63.89% protein and 1.23% lipid.) (Chenguang Biotechnology Group Kashgar Co., Ltd. Kashgar, China), Tapioca (Xiong Hebei Light Co., Ltd. Baoding, China), DL-methionine (Chuanheng Ecological Technology Co., Ltd. Deyang, China). ^2^ The premix provides minerals for a kilogram of diet: FeSO_4_·H_2_O (300 g/kg Fe), 10.00 g; MgSO_4_·H_2_O (160 g/kg Mg), 62.50 g; CuSO_4_·5H_2_O (250 g/kg Cu), 2.00 g; ZnSO_4_·H_2_O (340 g/kg Zn), 11.59 g; MnSO_4_·H_2_O (310 g/kg Mn), 3.77 g; CoCl_2_·6H_2_O (10 g/kg Co), 8.2 g; Ca (IO3)_2_ (30 g/kg I), 6.14 g; Na_2_SeO_3_ (10 g/kg Se), 20 g; KH_2_PO_4_ (520 g/kg K), 9.50 g; NaCl (390 g/kg Na), 7.63 g. All ingredients were diluted with CaCO_3_ to 1 kg. ^3^ The premix provides vitamins for kilogram of diet: retinyl acetate (500,000 IU/g), 8.89 g; thiamine (1000 g/kg), 1.40 g; riboflavine (800 g/kg), 2.92 g; pyridoxine (1000 g/kg), 1.47 g; folic acid (1000 g/kg), 0.13 g; cobalamine (10 g/kg), 6.67 g; Ascorbic acid (950 g/kg), 10.53 g; cholecalaiferol (500,000 IU/g), 0.16 g; DL-α-tocopherol acetate (500 g/kg), 14.67 g; menadione (1000 g/kg), 1.55 g; pantothenic acid (910 g/kg), 2.56 g; nicotinic acid (1000 g/kg), 2.51 g; biotin (20 g/kg), 3.33 g; inositol (980 g/kg), 27.21 g. All ingredients were diluted with corn starch to 1 kg. ^4^ Crude protein, lipid, ash, and moisture are expressed on a dry matter basis and given as means.

**Table 2 biology-14-00748-t002:** The primers used for qRT-PCR.

Item ^1^	Sequence	Tm (°C) ^2^	GenBank ID
IL-1β-QF	CATGTTGGATCGTTACGGCAT	61	XM_038733429.1
IL-1β-QR	CCCATCTTCACGTTTTAGGCAC		
TNF-α-QF	AGCAAGGAAGCAGACAACGG	59	XM_038723994.1
TNF-α-QR	ATTTGCCTCAATGTGTGACGAT		
IL-10-QF	AGCCAGCAGCATCATTACCA	59	XM_038696252.1
IL-10-QR	AGAACCAGGACGGACAGGAG		
TGF-β2-QF	ACACTTTGCTGAAACTGGGGA	64	XM_038710299.1
TGF-β2-QR	GAACACGGACGACACACAGGT		
*Sirt1*-QF	TACCAGAACAGCCACCAAGT	64.0	XM 038736812
*Sirt1*-QR	CATTATTACCAGCAGTCTCCGT		
*Yap*-QF	GCTTGTAAGGCCAACCTCCT	61.4	XM_038693048.1
*Yap*-QR	GATTTGGGGACCCTTGCGTA		
*Trf*-QF	GGGCAACAATCCCCAAACT	61.4	XM 038718037
*Trf*-QR	TCATCCACCAGACACTGAAAGG		
*Trfc*-QF	CTTCCTGTCGCCCTATGAGTC	64.5	XM 038718573
*Trfc*-QR	GTCTGCCTTAGGGTTGTTGGT		
*Gpx4*-QF	GTTTACGCATCCTTGCCTTCC	59.0	XM 038716292
*Gpx4*-QR	GCTCTTTCAGCCACTTCCACAA		
*Acsl4*-QF	GATCTGCACTCACCCCGACA	61.4	XM 038699899
*Acsl4*-QR	GCTCTGGACTCAAATGCACCT		
*Lpcat3*-QF	CAGCCCTTCTGGTATCGTTG	63.3	XM 038711111
*Lpcat3*-QR	ATACACCCTCCGCTATAACCC		
*Ptgs2*-QF	GCCTCGTCTGTAATAATGTCCG	64.5	XM 038715374
*Ptgs2*-QR	CTGAATGGGATGTGCTTGAGTT		
*β-actin*-QF	CCCCATCCACCATGAAGA	55.7	AF 253319.1
*β-actin*-QR	CCTGCTTGCTGATCCACAT		
*18S*-QF	TGAATACCGCAGCTAGGAATAATG	59.0	MH 018569.1
*18S*-QR	CCTCCGACTTTCGTTCTTGATT		

^1^ *Acsl4*: acyl-CoA Synthetase long chain family member 4; *β-actin:* beta actin; *Gpx4*: glutathione peroxidase 4; *Lpcat3*: lysophosphatidylcholine acyltransferase 3; *Ptgs2*: prostaglandin endoperoxide synthase 2; *Sirt1*: sirtuin 1; *18S:* 18S ribosomal RNA; *Trf*: transferrin; *Trfc:* transferrin receptor*; Yap*: yes associated protein; ^2^ Tm: melting temperature.

**Table 3 biology-14-00748-t003:** Growth performance of *M. salmoides* fed diets with different levels of CP for eight weeks.

Parameter ^1^	Diets						SEM	*p*-Value		
	FM	CP12	CP24	CP36	CP48	CP60		ANOVA	Linear	Quadratic
Initial body weight (g/fish)	39.40	39.37	39.27	39.47	39.40	39.27	0.093	0.609	0.476	0.509
Final body weight (g/fish)	99.77 ^a^	97.97 ^a^	99.37 ^a^	94.03 ^b^	92.21 ^bc^	89.79 ^c^	1.462	<0.001	0.000	0.193
Percent weight gain (%)	153.21 ^a^	148.87 ^a^	153.07 ^a^	138.25 ^b^	134.04 ^bc^	128.68 ^c^	4.732	<0.001	0.000	0.251
Specific growth rate (%)	1.47 ^a^	1.45 ^a^	1.47 ^a^	1.38 ^b^	1.35 ^bc^	1.31 ^c^	0.023	<0.001	0.000	0.224
Feed intake (g/fish)	63.51 ^a^	63.72 ^a^	61.28 ^ab^	60.94 ^ab^	58.37 ^b^	58.81 ^ab^	1.932	0.119	0.001	0.831
Feed efficiency (%)	95.03 ^ab^	92.11 ^ab^	98.25 ^a^	89.65 ^bc^	90.46 ^bc^	86.00 ^c^	3.287	0.061	0.027	0.383
Relative gut length (%)	78.03	77.56	78.42	77.69	77.67	76.32	1.673	0.738	0.256	0.342
Intestosomatic index (%)	0.84 ^a^	0.81 ^ab^	0.81 ^ab^	0.76 ^bc^	0.74 ^cd^	0.69 ^d^	0.037	0.001	<0.001	0.263
Viscerosomatic index (%)	8.48 ^a^	8.31 ^ab^	8.53 ^a^	7.94 ^b^	7.93 ^b^	7.46 ^c^	0.182	<0.001	<0.001	0.027
Hepatosomatic index (%)	2.33 ^a^	2.21 ^ab^	2.41 ^a^	2.08 ^bc^	1.95 ^c^	1.89 ^c^	0.126	<0.001	<0.001	0.104

^1^ Initial body weight (IBW, g/fish), final body weight (FBW, g/fish). Survival rate (SR, %) = 100 × final amount of fish/initial amount of fish; Percent weight gain (PWG, %) = 100 × (final body weight − initial body weight)/Initial body weight; Specific growth rate (SGR, %/d) = 100 × [ln (final body weight) − ln (initial body weight)]/days; Feed intake (FI, g/fish) = (feed offered in dry basis − uneaten feed in dry basis)/amount of fish; Feed efficiency (FE, %) = 100 × (final body weight − initial body weight)/Feed intake; Relative gut length (RGL, %) = 100 × intestine length/body length. Intestosomatic index (ISI, %) = 100 × wet intestine weight / wet body weight. Viscerosomatic index (VSI, %) = 100 × wet viscera weight/wet body weight; and Hepatosomatic index (HSI, %) = 100 × wet liver weight/wet body weight. *p*-value: significant probability associated with the F-statistic. Data presented as mean of three replicates. Different superscript letters (^a, b, c, d^) within a row indicate significant differences (*p* < 0.05). Without letter or bars shared the same letter represents no significant difference by Duncan’s test (*p* < 0.05, *n* = 12).

**Table 4 biology-14-00748-t004:** Effects of FM replacement by CP on flesh composition of *M. salmoides* (%, wet-basis).

Parameters ^1^	Diets						SEM	*p*-Value		
	FM	CP12	CP24	CP36	CP48	CP60		AVOVA	Linear	Quadratic
Moisture	70.95 ^c^	70.73 ^bc^	69.83 ^a^	70.01 ^a^	70.92 ^c^	70.50 ^ab^	0.015	0.039	0.596	0.025
Crude protein	18.50	18.67	19.00	19.00	18.83	19.00	<0.001	0.432	0.587	0.016
Crude lipid	7.09	7.17	7.25	7.15	7.00	7.06	<0.001	0.448	0.805	0.298
Ash	3.76	3.71	3.81	3.78	3.72	3.84	0.010	0.446	0.801	0.322
Protein production value	0.73	0.83	0.81	0.81	0.77	0.81	0.050	0.340	0.213	0.547
Lipid production value	2.30	2.50	2.45	2.44	2.23	2.48	0.110	0.273	0.306	0.926

^1^ PPV: protein production value; LPV: lipid production value. p-value: significant probability associated with the F-statistic. Data presented as mean of three replicates. Different superscript letters (^a, b, c^) within a row indicate significant differences (*p* < 0.05). Without letter or bars shared the same letter represents no significant difference by Duncan’s test (*p* < 0.05, *n* = 9).

**Table 5 biology-14-00748-t005:** Effects of FM replacement by CP on serum and liver biochemical indices of *M. salmoides*.

Parameters ^1^	Diets						SEM	*p*-Value		
	FM	CP12	CP24	CP36	CP48	CP60		ANOVA	Linear	Quadratic
Serum										
Triglyceride (mmol/L)	9.19 ^a^	8.46 ^a^	14.99 ^c^	17.75 ^d^	17.78 ^d^	12.30 ^b^	3.462	<0.001	0.000	0.000
Total cholesterol (mmol/L)	11.41 ^ab^	10.78 ^a^	12.03 ^b^	13.07 ^c^	13.19 ^c^	12.82 ^c^	0.353	<0.001	0.000	0.092
Alkalinephosphatase (King /g prot)	7.70 ^b^	7.74 ^b^	7.66 ^b^	9.52 ^c^	7.36 ^ab^	6.39 ^a^	0.493	<0.001	0.067	0.001
L-Hydroxyproline (ug/mL)	5.90 ^a^	9.12 ^b^	8.88 ^b^	9.37 ^bc^	13.32 ^d^	11.26 ^c^	0.996	<0.001	0.000	0.133
Liver										
Aspartate aminotransferase (U/g prot)	39.05 ^b^	36.43 ^b^	35.87 ^ab^	32.41 ^a^	39.65 ^b^	53.19 ^c^	1.983	<0.001	0.000	0.000
Alanine aminotransferase (U/g prot)	160.04 ^b^	163.35 ^b^	161.71 ^b^	141.58 ^a^	168.07 ^b^	195.69 ^c^	8.372	<0.001	0.000	0.000

^1^ TG: triglyceride; TC: total cholesterol; AST: aspartate aminotransferase; ALT: alanine aminotransferase; AKP: alkalinephosphatase; HYP: L-Hydroxyproline. *p*-value: significant probability associated with the F-statistic. Data presented as mean of three replicates. Different superscript letters (^a, b, c, d^) within a row indicate significant differences (*p* <0.05). Without letter or bars shared the same letter represents no significant difference by Duncan’s test (*p* < 0.05, *n* = 9).

## Data Availability

The datasets used and analyzed during the current study are available from the corresponding author on reasonable request.

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
