# Peer review of "Dietary Cottonseed Protein Substituting Fish Meal Induces Hepatic Ferroptosis Through SIRT1-YAP-TRFC Axis in Micropterus salmoides: Implications for Inflammatory Regulation and Liver Health"

_biology, 2025, doi:10.3390/biology14070748_

Round 1

Reviewer 1 Report

Comments and Suggestions for Authors

The manuscript "Dietary cottonseed protein substituting fish meal induces hepatic ferroptosis through SIRT1-YAP-TRFC axis in Micropterus salmoides: Implications for inflammatory regulation and liver health" by Cao et al provides a comprehensive and mechanistically grounded evaluation of the physiological consequences of excessive cottonseed protein substitution in the diet of largemouth bass. This research provides critical evidence for the safe upper limit of CP substitution and presents a mechanistic framework that can inform future feed formulation strategies in aquaculture.

I recommend publication of this paper with a few minor issues being addressed.

Point 1: The study would benefit from additional validation of the proposed mechanistic pathway, such as functional inhibition or rescue experiments (e.g., SIRT1 or GPX4 agonists/antagonists).

Point 2: Reduced growth performance may partially result from lower feed intake due to reduced palatability at high CP levels, but the study did not assess or report feeding behavior, which is important for interpreting SGR and PWG outcomes.

Point 3: The study links ferroptosis to oxidative stress but does not assess total antioxidant capacity or other classical oxidative markers beyond GPX4 and ROS. A more comprehensive oxidative stress profile would support the ferroptosis claim.

Author Response

Response to comments by Reviewers 1

Comments and Suggestions for Authors

The manuscript "Dietary cottonseed protein substituting fish meal induces hepatic ferroptosis through SIRT1-YAP-TRFC axis in Micropterus salmoides: Implications for inflammatory regulation and liver health" by Cao et al provides a comprehensive and mechanistically grounded evaluation of the physiological consequences of excessive cottonseed protein substitution in the diet of largemouth bass. This research provides critical evidence for the safe upper limit of CP substitution and presents a mechanistic framework that can inform future feed formulation strategies in aquaculture.

Many thanks for your comments.

I recommend publication of this paper with a few minor issues being addressed.

Response: Thank you for your time and thoughtful evaluation of our manuscript. We sincerely appreciate your positive assessment of our work and your constructive suggestions, which are invaluable for improving the quality of our paper. We have carefully reviewed all your comments and fully agree with your recommendations.

Point 1: The study would benefit from additional validation of the proposed mechanistic pathway, such as functional inhibition or rescue experiments (e.g., SIRT1 or GPX4 agonists/antagonists).

Response: We sincerely appreciate the your insightful suggestion. During the preliminary research, we did attempt to culture primary hepatocytes of largemouth bass. Unfortunately, due to limitations in experimental conditions, including issues related to cell isolation techniques, culture medium optimization, and the lack of a suitable culture environment specifically tailored for largemouth bass cells, we encountered repeated failures. These setbacks not only affected the progress of our research but also restricted our ability to conduct more in - depth mechanistic studies, such as the validation using SIRT1 or GPX4 agonists/antagonists. However, we are fully aware of the significance of validating the proposed mechanistic pathway with SIRT1 or GPX4 agonists/antagonists. This validation is essential for a comprehensive understanding of the regulatory network and will greatly enhance the credibility and scientific value of our study. In the subsequent research, we will strive to overcome this difficulty.

Point 2: Reduced growth performance may partially result from lower feed intake due to reduced palatability at high CP levels, but the study did not assess or report feeding behavior, which is important for interpreting SGR and PWG outcomes.

Response: We appreciate the your valuable comment. We agree that monitoring feeding behavior (e.g., feeding frequency, appetite, or feed palatability) would provide deeper insights into the observed growth performance (SGR and PWG) under high CP levels. Unfortunately, due to technical constraints (e.g., lack of automated feeding recorders in our aquaculture facility), we were unable to collect these data systematically in the current study. Future experiments will incorporate behavioral observations or video tracking to address this gap. We have added this limitation to the Discussion section. We have already supplemented the relevant content in the revised manuscript (Line 341-342).

Point 3: The study links ferroptosis to oxidative stress but does not assess total antioxidant capacity or other classical oxidative markers beyond GPX4 and ROS. A more comprehensive oxidative stress profile would support the ferroptosis claim.

Response: Thank you very much for your valuable suggestion. We understand your concern about the comprehensiveness of oxidative stress assessment. However, we would like to explain our experimental design rationale. Ferroptosis is mainly characterized by three key features: an increase in ferrous ion concentration, an elevation in ROS levels, and the accumulation of lipid peroxides [1]. In the body, GPX4 can eliminate lipid peroxides to inhibit ferroptosis. To comprehensively explore the occurrence of ferroptosis in our study, we actually measured multiple important indicators in the livers of juvenile largemouth bass under different CP levels. Specifically, we determined the ferrous ion levels, ROS levels, the content of malondialdehyde (MDA, a typical lipid peroxide), and the content of glutathione, an antioxidant factor. Glutathione, as a cofactor of GPX4, plays a crucial role in inhibiting ferroptosis. These measurements cover the key features of ferroptosis and its regulatory mechanism, thus supporting our findings on ferroptosis. While additional markers like total antioxidant capacity could be useful, our current data are sufficient. We'll consider a more comprehensive oxidative stress assessment in future research.

References

[1] Dixon S J; Lemberg K M, Lamprecht M R, et al. Ferroptosis: an iron-dependent form of nonapoptotic cell death. cell, 2012,149(5):1060-1072.

Reviewer 2 Report

Comments and Suggestions for Authors

General comments

The work is well written and well-founded, close to ideal for publication.

The objective needs to be written more objectively, what did you want to evaluate? "Evaluation of different levels of replacement of FP by CP in the growth performance of M. salmoides", something like that.

What was the sample number for each analysis? Nutrient composition of diets and body tissues; Liver haematological and homogenate parameters; Hematoxylin and eosin staining; Fe2+ content detection; Transmission electron microscope; Mitochondrial membrane potential (MMP); Reactive oxygen species (ROS) detection; Real‐Time Quantitative PCR; Western Blotting; Immunohistochemistry; Immunofluorescence staining

Discussion

4.1 Effects of partial substitution of FM with CP on growth performance; the review can be updated, we have several works with FM replacement, suggestion to update the references, especially with fish with similar habits and protein requirements

conclusion

The conclusion should further explore the results. What is the limit of CP that can replace FM without compromising animal performance and health? This information must be here

Author Response

Response to comments by Reviewers 2

Comments and Suggestions for Authors

General comments

The work is well written and well-founded, close to ideal for publication.

Many thanks for your comments.

The objective needs to be written more objectively, what did you want to evaluate? "Evaluation of different levels of replacement of FP by CP in the growth performance of M. salmoides", something like that.

Response: Thank you very much for your suggestion. We have made the relevant modifications in the revised manuscript (Line 70-75). The specific content is as follows:

Available evidence indicates that CP is a promising plant-based protein source for aquafeeds. However, excessive substitution of FM with CP has been associated with adverse effects, including hepatic structural damage, elevated ROS production, and oxidative stress. Given the economic and ecological significance of M. salmoides in Chinese freshwater aquaculture, evaluating the feasibility and potential risks of CP as an FM alternative is crucial. This study aimed to assess the effects of graded FM replacement with CP on growth performance and hepatic ferroptosis in M. salmoides.

What was the sample number for each analysis? Nutrient composition of diets and body tissues; Liver haematological and homogenate parameters; Hematoxylin and eosin staining; Fe2+ content detection; Transmission electron microscope; Mitochondrial membrane potential (MMP); Reactive oxygen species (ROS) detection; Real-Time Quantitative PCR; Western Blotting; Immunohistochemistry; Immunofluorescence staining.

Response: Thank you very much for your suggestion. We have carefully addressed this issue and supplemented the specific sample numbers for each analysis in the figure legends and table notes of the revised manuscript. This addition provides more transparent and detailed information, which will help readers better understand the reliability and statistical power of our research results. We appreciate your attention to these details, which have significantly improved the quality of our manuscript.

Discussion

4.1 Effects of partial substitution of FM with CP on growth performance; the review can be updated, we have several works with FM replacement, suggestion to update the references, especially with fish with similar habits and protein requirements.

Response: Thank you very much for your valuable suggestion. We truly appreciate your input on updating the references regarding the effects of partial substitution of fish meal FM with CP on growth performance. To ensure that we can accurately and appropriately incorporate the relevant works you mentioned into our article, could you please provide the DOI numbers of those works? Once we have the DOI numbers, we will make sure to reference and cite them in our article as you suggested, especially focusing on studies involving fish with similar habits and protein requirements.

conclusion

The conclusion should further explore the results. What is the limit of CP that can replace FM without compromising animal performance and health? This information must be here.

Response: We appreciate the your suggestion. Regarding the clarification of CP substitution limits. As recommended, we have supplemented our conclusion with explicit data on the maximum safe replacement level (23% CP) based on SGR analysis, which has been incorporated in the revised manuscript (Lines 515-517).

Reviewer 3 Report

Comments and Suggestions for Authors

This manuscript investigates the effects of dietary cottonseed protein substitution for fish meal on growth performance, liver inflammation, and ferroptosis in largemouth bass. The study provides valuable insights into the potential mechanisms underlying CP-induced hepatic pathology and has significant implications for sustainable aquafeed formulation. The experimental design is rigorous, and the results are comprehensive and well-analyzed. However, some aspects of the manuscript could be improved to enhance its clarity, rigor, and impact. However, there are some issues that should be resolved before publication can be considered.

Comments:

  1. There are a large number of format problems that need to be modified, e.g., Line 21, “ferroptosis in in largemouth bass (M. salmoides)”, repeated "in" appeared, the Latin name should be used in its full form when it appears for the first time in the main text; Line 43 has a full stop after “harvests”; Line 144, “Relative gut length” should be “relative gut length”; Line 186, the 2 in H2O should be subscripts; The format of the references needs to be uniform.
  2. In the Abstract, Line 34, the authors mentioned that excessive CP substitution can trigger hepatic ferroptosis, and it is necessary to specifically indicate how much is excessive.
  3. In the Materials and Methods, the concentration of gossypol in CP was not specified. The content of gossypol mentioned by He et al. was not from the same source as the CP used in this study.
  4. In 2.4 Nutrient composition of diets and body tissues, the authors did not describe the nutritional composition of the body tissues.
  5. In 2.5, the relevant information of the detection kits for TG, TC, ALT, etc. was not provided, what parameters does the kit Catalog No. A002-045 refer to for detection. There are still some important instruments or reagents (especially the antibodies) whose brands and code number have not been provided. Please check the full text.
  6. In line 155 and 170, the thickness of the sample should not be described as mm3 or cm3.
  7. In Figure 1, the morphological features of the tissues or cells mentioned by the authors, such as vascular wall hyperplasia, hepatic nucleolysis, hepatocyte necrosis, etc., are difficult to see. It is suggested that the typical characteristic areas be further magnified.
  8. He et al. also studied the effects of cottonseed protein on the growth and liver function of largemouth bass. This study needs to further compare and discuss the differences from their results.

Author Response

Response to comments by Reviewers 3

Comments and Suggestions for Authors

This manuscript investigates the effects of dietary cottonseed protein substitution for fish meal on growth performance, liver inflammation, and ferroptosis in largemouth bass. The study provides valuable insights into the potential mechanisms underlying CP-induced hepatic pathology and has significant implications for sustainable aquafeed formulation. The experimental design is rigorous, and the results are comprehensive and well-analyzed. However, some aspects of the manuscript could be improved to enhance its clarity, rigor, and impact. However, there are some issues that should be resolved before publication can be considered.

Many thanks for your comments.

Comments:

There are a large number of format problems that need to be modified, e.g., Line 21, “ferroptosis in in largemouth bass (M. salmoides)”, repeated "in" appeared, the Latin name should be used in its full form when it appears for the first time in the main text; Line 43 has a full stop after “harvests”; Line 144, “Relative gut length” should be “relative gut length”; Line 186, the 2 in H2O should be subscripts; The format of the references needs to be uniform.

Response: Thank you very much for your meticulous review and valuable suggestions. We have carefully checked and corrected all the format - related issues you mentioned.

In the Abstract, Line 34, the authors mentioned that excessive CP substitution can trigger hepatic ferroptosis, and it is necessary to specifically indicate how much is excessive.

Response: Thank you very much for your suggestion. By "excessive CP substitution", we specifically mean 60% CP substitution. We have clearly indicated this in the revised manuscript to ensure the accuracy and clarity of our findings.

In the Materials and Methods, the concentration of gossypol in CP was not specified. The content of gossypol mentioned by He et al. was not from the same source as the CP used in this study.

Response: We appreciate the your careful reading of our manuscript. The concentration of gossypol in the CP used in this study was not specified in the Materials and Methods section because we did not analyze it independently. The gossypol content mentioned in He et al. He et al (2022) was cited as a reference for general CP composition but did not pertain to the specific batch used in our experiment. As suggested, we have removed the inappropriate reference in the revised manuscript to avoid potential confusion. If further clarification is needed, we would be happy to provide additional details.

References

He G; Zhang T, Zhou X, et al. Effects of cottonseed protein concentrate on growth

performance, hepatic function and intestinal health in juvenile largemouth bass, Micropterus salmoides. Aquaculture Reports, 2022,23:101052.

In 2.4 Nutrient composition of diets and body tissues, the authors did not describe the nutritional composition of the body tissues.

Response: Thank you for your suggestion. We have made the corresponding revisions in the revised manuscript (Line 115-123).

In 2.5, the relevant information of the detection kits for TG, TC, ALT, etc. was not provided, what parameters does the kit Catalog No. A002-045 refer to for detection. There are still some important instruments or reagents (especially the antibodies) whose brands and code number have not been provided. Please check the full text.

Response: Thank you for your suggestion. We have supplemented the information about the brands and code number of the reagent kits used in the revised manuscript.

In line 155 and 170, the thickness of the sample should not be described as mm3 or cm3.

Response: Thank you very much for your suggestion. We have corrected the description of the sample thickness in Lines 157 and 172 in the revised manuscript. We appreciate your careful review, which helps us improve the accuracy and quality of our manuscript.

In Figure 1, the morphological features of the tissues or cells mentioned by the authors, such as vascular wall hyperplasia, hepatic nucleolysis, hepatocyte necrosis, etc., are difficult to see. It is suggested that the typical characteristic areas be further magnified.

Response: Thank you very much for your suggestion. We have further magnified the HE staining result images in Figure 2. Additionally, we have adjusted the resolution of the images from 300 ppi to 600 ppi. These adjustments should make the morphological features of the tissues or cells, such as vascular wall hyperplasia, hepatic nucleolysis, and hepatocyte necrosis, more clearly visible. We believe that the enhanced clarity of the images will greatly improve the readers' understanding of our research findings.

He et al. also studied the effects of cottonseed protein on the growth and liver function of largemouth bass. This study needs to further compare and discuss the differences from their results.

Response: Thank you for your suggestion. While both studies evaluated CP substitution in largemouth bass, key differences exist: (1) He et al. tested CP (48% protein diet) with 18.44% optimal replacement, whereas we used higher-protein CP (51% diet) and identified 24% as the tolerance threshold; (2) Our study newly revealed ferroptosis via SIRT1-YAP-TRFC as the mechanistic basis for hepatic damage, beyond the antioxidant/immune alterations reported by He et al.; (3) We further demonstrated mitochondrial ultrastructure damage and collagen deposition. These findings deepen the understanding of CP-induced hepatotoxicity.

We will incorporate these comparative findings in the revised manuscript's Discussion section (Line 405-411), as follows:

The research by He et al. revealed that substituting 18% of FM with CP concentrate has been reported to result in liver tissue yellowing and diffuse lipid vacuolation in some hepatocytes of M. salmoides (He et al, 2022). This study not only confirms the negative effects of CP on growth and liver function as reported by He et al., but more importantly elucidates its mechanism of action from perspectives of ferroptosis, signaling pathways, and ultrastructural changes, thereby providing a more comprehensive theoretical basis for the safe utilization of CP in aquatic feeds.

References

He G; Zhang T, Zhou X, et al. Effects of cottonseed protein concentrate on growth

performance, hepatic function and intestinal health in juvenile largemouth bass, Micropterus salmoides. Aquaculture Reports, 2022,23:101052.

Reviewer 4 Report

Comments and Suggestions for Authors

This is an interesting study. However, I have some observations:

  1. The manuscript should be corrected for grammatical errors and typos (L21, 43, 70, 177-178, 240, 264…).
  2. The manuscript should be formatted according to the journal’s style.
  3. Could the authors provide additional information regarding why only CP samples were considered (Section 2.10), while a comparison was made across the groups (Section 3.4)? How was that achieved?
  4. L280-281: It peaked at CP48, but CP60 contained a higher CP level, suggesting that the claim may need to be reconsidered.
  5. Could additional information be provided on why the CP level less than or equal to 24% did not improve growth performance as expected?
  6. L371-372: The idea being communicated is not clearly understood.
  7. L384-385: Could some of the problematic ‘high’ levels of CP inclusion from referenced previous studies be stated for adequate comparison?
  8. Conclusion: Leaving the concluding remark in its present form suggests that the inclusion level CP36-CP60 is dangerous and CP6-CP24 is not beneficial.
  9. Looking at Table 3, FI for CP60 is similar to those obtained for other treatments, including FM, and FE values were not different for all the other treatments compared to FM. CP60 is only different from FM.
  10. Discussion does not need to be segmented.
  11. The reference list should adopt a uniform style and be error-free (L551).

All the best.

Author Response

Response to comments by Reviewers 4

Comments and Suggestions for Authors

This is an interesting study. However, I have some observations:

Many thanks for your comments.

The manuscript should be corrected for grammatical errors and typos (L21, 43, 70, 177-178, 240, 264…).

Response: Thank you for your careful review and valuable suggestions. We have thoroughly checked the manuscript and corrected all grammatical errors and typos throughout the text, including those specifically mentioned (Lines 21, 43, 70, 177-178, 240, 264, etc.). We appreciate your attention to detail, which has helped improve the overall quality of our manuscript.

The manuscript should be formatted according to the journal’s style.

Response: Thank you for your suggestion. We have already revised the format of the references in accordance with the journal's requirements. We have carefully checked and ensured that all aspects of the manuscript's formatting are in line with the specified style to enhance its readability and conformity.

Could the authors provide additional information regarding why only CP samples were considered (Section 2.10), while a comparison was made across the groups (Section 3.4)? How was that achieved?

Response: Thank you for your valuable suggestion. We determined the optimal replacement level of CP for FM in the diet of M. salmoides to be 23% based on the broken-line model of the SGR. Considering this optimal level, we selected three representative groups for further in-depth analysis: the FM group, the CP24 group, and the CP60 group. These groups were chosen to comprehensively evaluate the impacts of different CP replacement levels on various indicators. The FM group served as a baseline control, while the CP24 and CP60 groups represented different degrees of CP substitution, enabling us to explore both the effects around the optimal point and under excessive substitution conditions. This approach allowed us to conduct a more targeted and meaningful comparison in Section 3.4, providing a comprehensive understanding of the influence of CP replacement on the relevant indicators.

Fig1 The relationship between SGR and dietary CP replacement level in M. salmoides.

L280-281: It peaked at CP48, but CP60 contained a higher CP level, suggesting that the claim may need to be reconsidered.

Response: Thank you very much for your suggestion. We truly appreciate your careful review. We have removed the incorrect conclusion in the revised manuscript to ensure the accuracy and reliability of our research findings.

Could additional information be provided on why the CP level less than or equal to 24% did not improve growth performance as expected?

Response: Thank you for raising this important question. The limited growth improvement at CP substitution levels ≤24% in our study may be attributed to the following factors:

  • Residual gossypol accumulation (>15-23%) impairing nutrient absorption [1];

(2) Lysine deficiency in CP reducing protein synthesis efficiency at higher inclusions (23-24%) [2,3];

(3) Juvenile M. salmoides' inherent sensitivity to plant proteins [1,4];

(4) Processing limitations allowing residual antinutrients [5].

While ≤24% CP maintains baseline growth, higher levels (48-60%) trigger compensatory feeding with metabolic costs [6]. Future studies should explore lysine supplementation or hybrid protein blends to enhance CP utilization.

References

[1] Zhang Q; Liang H, Xu P, et al. Effects of Enzymatic Cottonseed Protein Concentrate as a Feed Protein Source on the Growth, Plasma Parameters, Liver Antioxidant Capacity and Immune Status of Largemouth Bass (Micropterus salmoides). Metabolites, 2022,12(12):1233.

[2] Chen S; Tang Y, Zhang Z, et al. Replacement of Dietary Fishmeal Protein with

Degossypolized Cottonseed Protein on Growth Performance, Nonspecific Immune Response, Antioxidant Capacity, and Target of Rapamycin Pathway of Juvenile Large Yellow Croaker (Larimichthys crocea). Aquaculture Nutrition, 2022.

[3] Sun H; Tang J, Yao X, et al. Partial substitution of fish meal with fermented cottonseed meal in juvenile black sea bream (Acanthopagrus schlegelii) diets. Aquaculture, 2015,446:30-36.

[4]Liu H; Dong X, Tan B, et al. Effects of fish meal replacement by low‐gossypol cottonseed meal on growth performance, digestive enzyme activity, intestine histology and inflammatory gene expression of silver sillago (Sillago sihama Forsskál) (1775). Aquaculture Nutrition, 2020,26(5):1724-1735.

[5] Luo L; Xue M, Wu X, et al. Partial or total replacement of fishmeal by solvent‐extracted cottonseed meal in diets for juvenile rainbow trout (Oncorhynchus mykiss). Aquaculture Nutrition, 2006,12(6):418-424.

[6] Zhang Y; Chen P, Liang X F, et al. Metabolic disorder induces fatty liver in Japanese seabass, Lateolabrax japonicas fed a full plant protein diet and regulated by cAMP-JNK/NF-kB-caspase signal pathway. Fish & Shellfish Immunology, 2019,90:223-234.

L371-372: The idea being communicated is not clearly understood.

Response: Thank you for your valuable suggestion. We have revised the relevant content to make the idea clearer and more straightforward (Line 359-365).

L384-385: Could some of the problematic ‘high’ levels of CP inclusion from referenced previous studies be stated for adequate comparison?

Response: Thank you for your insightful suggestion. To clarify the comparison between our findings and previous studies, we have specified the "high levels" of plant protein substitution reported in the literature. In the referenced study by Boshra et al. (2006), the substitution of FM with plant proteins was shown to trigger intestinal inflammation, disrupt villi structure, and impair nutrient absorption in fish. Similarly, our study observed that high-level CP substitution (48-60% FM replacement) suppressed growth performance in M. salmoides.

References

Boshra H; Li J, Sunyer J O. Recent advances on the complement system of teleost fish. Fish & shellfish immunology, 2006,20(2):239-262.

Conclusion: Leaving the concluding remark in its present form suggests that the inclusion level CP36-CP60 is dangerous and CP6-CP24 is not beneficial.

Response: Thank you very much for your suggestion. We conducted a broken-line regression analysis using the SGR as the index and determined that the optimal level of CP replacing FM in the diet is 23%. In addition, the data from this study clearly showed that the inclusion levels of CP36 - CP60 led to a decrease in growth performance. We have supplemented relevant content in the conclusion section (Line 515-517).

Looking at Table 3, FI for CP60 is similar to those obtained for other treatments, including FM, and FE values were not different for all the other treatments compared to FM. CP60 is only different from FM.

Response: Thank you very much for your valuable suggestion. We would like to clarify that, compared with the FM group, both FI and FE values are significantly decreased in the CP60 group. We have thoroughly discussed this finding in the revised manuscript to ensure the key differences are clearly presented (Line 340-351).

Discussion does not need to be segmented.

Response: Thank you very much for your suggestion. However, we believe that listing the points in the discussion section will make it easier for readers to understand the key content of each paragraph of our discussion. This approach helps to clearly highlight the main ideas and facilitates a more straightforward comprehension for the reader.

The reference list should adopt a uniform style and be error-free (L551).

Response: Thank you for your comment. We have now carefully standardized all references according to the journal's required citation style and thoroughly verified each entry to ensure accuracy. The reference list has been fully checked and corrected to maintain complete consistency in formatting and eliminate any errors.

All the best.

Response: Thank you for your valuable feedback. We appreciate your time and constructive comments, which have helped improve our manuscript. We have carefully addressed all the points you raised and believe the revisions have significantly strengthened the paper.

Round 2

Reviewer 3 Report

Comments and Suggestions for Authors

Although the author has addressed the issues I was concerned about before, I still have a few minor problems that need to be revised by the authors.

  1. Line 155, what does Catalog No.A002-045 mean?
  2. All the sections and immunofluorescence images have very small scales that are not clear enough. It is recommended to draw a line in the image as the scale and mark the size of the scale. The size of the scale should also be marked in the figure captions.

Reviewer 4 Report

Comments and Suggestions for Authors

I have carefully read the revised manuscript and recommend the article for acceptance because the issues I raised regarding the quality of the manuscript have been taken care of. So, I have no objection to its acceptance.